# Facilitating Multimodal Classification via Dynamically Learning Modality Gap

**Yang Yang**[†]**, Fengqiang Wan**[†]**, Qing-Yuan Jiang**[∗†]**, Yi Xu**[‡]
[†]Nanjing University of Science and Technology
[‡]Dalian University of Technology
`{yyang,fqwan,jiangqy}@njust.edu.cn, yxu@dlut.edu.cn`

## Abstract

Multimodal learning falls into the trap of the optimization dilemma due to the modality imbalance phenomenon, leading to unsatisfactory performance in real applications. A core reason for modality imbalance is that the models of each modality converge at different rates. Many attempts naturally focus on adjusting learning procedures adaptively. Essentially, the reason why models converge at different rates is because the difficulty of fitting category labels is inconsistent for each modality during learning. From the perspective of fitting labels, we find that appropriate positive intervention label fitting can correct this difference in learning ability. By exploiting the ability of contrastive learning to intervene in the learning of category label fitting, we propose a novel multimodal learning approach that dynamically integrates unsupervised contrastive learning and supervised multimodal learning to address the modality imbalance problem. We find that a simple yet heuristic integration strategy can significantly alleviate the modality imbalance phenomenon. Moreover, we design a learning-based integration strategy to integrate two losses dynamically, further improving the performance. Experiments on widely used datasets demonstrate the superiority of our method compared with state-of-the-art (SOTA) multimodal learning approaches. The code is available at `https://github.com/njustkmg/NeurIPS24-LFM`.

## 1 Introduction

Multimodal learning (MML) [3, 38, 47, 41, 10, 26, 31, 43, 13] integrates heterogeneous information from different modalities to build an effective way to perceive the world. Over the past decades, multimodal learning has made incredible progress [31, 10, 13] and become a hot research topic with a wide range of real applications including image caption [6, 14], cross-modal retrieval [21, 44, 54, 37], vision reasoning [32, 8], action recognition [23, 28], and so on.

In multimodal learning, several recent studies [38, 31] have revealed an interesting phenomenon, i.e., the performance of the multimodal model is far from the upper bound or even inferior to the unimodal in certain situations. The root of this problem lies in the existence of the modality imbalance phenomenon [38]. Concretely, there commonly exists dominant modality and non-dominant modality in heterogeneous multimodal data. Multimodal learning usually adopts a uniform objective. Due to greediness [42], the optimization tends to dominant modality while neglecting the non-dominant one during joint training, thus leading to unsatisfactory performance in real applications.

Recently, many impressive works [38, 11, 31, 13, 24] have been proposed to address the modality imbalance problem. Early pioneering approaches, such as gradient blending (G-Blend) [38], on-the-fly

---

[∗]Corresponding author.

gradient modulation (OGM) [31], adaptive gradient modulation (AGM) [24], and prototypical modality rebalance (PMR) [13], focus on designing customized learning strategies for different modalities to adjust the optimization of dominant and non-dominant modality. These methods demonstrate that suppressing the optimization of the dominant modality can alleviate the modality imbalance problem to a certain extent. Besides, several attempts, including uni-modal teachers (UMT) [11] and balanced multimodal learning [42], try to introduce extra networks as an auxiliary module to facilitate multimodal learning.

Although the aforementioned approaches can boost performance in MML, these solutions are based on the phenomenon of inconsistent learning speed itself and do not study the underlying causes of modality imbalance. We can't help but ask what is the essential reason behind this phenomenon. Is there a bias in the process of fitting category labels for different modalities? We carry out a simple experiment on Kinetic-sSounds dataset to seek answers. We adopt two types of labels to explore the influence of fitting labels. The first type is one-hot labels which indicate the category of each sample, where the loss is denoted as $L_S$. The second type is label free, i.e., uniform label $1/c$ for all samples, where $c$ denotes the number of categories. The second loss is defined as $L_U$. Furthermore, we define a mixed loss $0.7L_S + 0.3L_U$, which is actually the label smoothing [35] strategy, by combining one-hot labels and uniform labels. The accuracy is reported in Figure 1. From Fig-

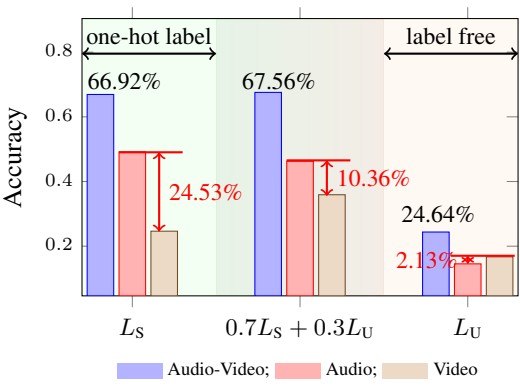

Figure 1: The influence of labels fitting on performance gaps (best view in color), where $L_S$ and $L_U$ denote the loss with one-hot labels and uniform labels (label free).

ure 1, we can observe that with proper intervention by using uniform labels, the performance is slightly better than the model that fits one-hot labels. More importantly, the performance gap becomes smaller if we learn from uniform labels. This means that the difference between audio and video modalities is smaller in feature space when we reduce the weight of one-hot labels. The experiment inspires us that appropriate intervention label fitting can alleviate the difference in the learning ability of different modalities. In this way, we can reduce the modality performance gap and further mitigate the modality imbalance phenomenon as the smaller the difference in modality performance, the less severe the modality imbalance [39, 20, 52]. This also implies that fitting category labels is a core cause of modality imbalance in multimodal learning.

How do we impose positive intervention in multimodal learning so that the impact of fitting labels on modality imbalance is as low as possible without affecting the overall performance? For multimodal learning, although the models are learned from the heterogeneous data, we hope that multimodal data describing the same entity should be as close as possible in the feature space, which is usually modeled as contrastive learning [33]. Contrastive learning aims to learn similar representations for data pairs of different modalities [12], thus aligning the multimodal representations for all modalities. Ideally, contrastive learning can also mitigate the effect of modality imbalance problem. Hence, we introduce contrastive learning to impose positive intervention in multimodal learning to alleviate the impact of fitting labels.

In this paper, we propose a novel multimodal learning approach by integrating unsupervised contrastive learning and supervised multimodal learning dynamically. Specifically, after demonstrating the effectiveness of unsupervised contrastive learning in multimodal learning, we design two dynamical integration strategies, i.e., a heuristic and a learning-based integration strategy. Our contributions are outlined as follows: (1). We observe a key phenomenon: fitting category labels leads to a larger performance gap between different modalities. To the best of our knowledge, this is the first time that the modality imbalance problem has been analyzed from the perspective of category label fitting. (2). We propose a novel multimodal learning approach by integrating unsupervised contrastive learning and supervised multimodal learning. Two strategies are designed for dynamic integration. (3). Extensive experiments on widely used datasets show that our proposed approach can significantly outperform other baselines to achieve state-of-the-art performance.

## 2 Related Work

### 2.1 Multimodal Learning

Multimodal learning aims to leverage multimodal data from different sources to improve model performance. Based on the fusion strategy, multimodal learning approaches can be categorized into early fusion [38, 46, 52, 50], late fusion [48, 47, 1, 27], and hybrid fusion [22, 53]. Early fusion methods aim to integrate multimodal features to study the interrelationship between different modalities with joint representations when features are extracted by encoders. Representative early fusion methods include G-Blend [38], association-based fusion (AF) [26], and DOMFN [46]. On the contrary, late fusion methods leverage the prediction of each model to make final decisions. Late fusion methods can be divided into two categories, i.e., soft late fusion and hard late fusion, where the former utilizes the confidence score to make decisions and the latter the category decision of each model. Pioneering late fusion methods include modality-specific learning rate (MSLR) [47]. Hybrid fusion methods try to amalgamate the advantages of both early and late fusion methods. Representative hybrid fusion methods include multimodal transfer module (MMTM) [22] and balanced multi-modal learning [42]. Although these methods explore the algorithms and applications in multimodal learning, all of them assume that each modality can make sufficient contributions to achieve satisfactory performance during the training procedure.

### 2.2 Imbalanced Multimodal Learning

In reality, an obvious situation is that multimodal data and models are diverse, which naturally leads to different contributions during the training procedure. Recent works [38, 31, 13, 24] have shown that modality imbalance is a ubiquitous phenomenon and often results in unsatisfactory performance or even worse than unimodal algorithms in some cases. Considering the existence of dominant modality and non-dominant modality, early pioneering approaches [38, 31, 13] focus on adjusting learning speed for different modalities with customized learning strategies to balance the optimization of dominant and non-dominant modality. For instance, G-Blend [38] proposes to minimize the overfitting-to-generalization ratio (OGR) by using a gradient blending technique based on the modality's overfitting behavior. OGM [31] utilizes an on-the-fly gradient modulation strategy to control the modality's optimization procedure. To achieve the purpose of balanced multimodal learning, PMR [13] designs a prototypical modal rebalance strategy to facilitate the learning of non-dominant modality. Other attempts [11, 42] try to utilize extra networks to facilitate multimodal learning. Concretely, UMT [11] utilizes the teacher networks to distill the pretrained unimodal features to the multimodal network to tackle the modality imbalance problem. Balanced multimodal learning [42] utilizes the gradient norm and model parameters' norm to define conditional learning speed and uses it to guide the learning procedure. These methods alleviate the modality imbalance problem to a certain extent.

## 3 Methodology

We present our proposed method in this section. Specifically, we first present the problem definition of multimodal learning. Then, we introduce unsupervised contrastive learning to impose positive intervention in multimodal learning and propose two dynamical integration strategies to maximize the learning collaboration of unsupervised contrastive learning and supervised multimodal learning.

### 3.1 Preliminary

For the sake of simplicity, we use boldface lowercase letters like $\boldsymbol{a}$ and boldface uppercase letters like $\boldsymbol{A}$ to denote vectors and tensors, respectively. The $i$-th element of $\boldsymbol{a}$ is denoted as $a_i$. Furthermore, we use $\|\cdot\|_2$ to denote $L_2$ norm of the vectors.

The goal of multimodal learning is to train a model to predict the category labels for given multimodal data. Without any loss of generality, we use $\boldsymbol{X} = \{\boldsymbol{x}_i\}_{i=1}^n$ to denote the training data points, where each data point is with $m$ modalities, i.e., $\boldsymbol{x}_i = \{\boldsymbol{x}_i^{(j)}\}_{j=1}^m$. The category labels are represented as $\boldsymbol{Y} = \{\boldsymbol{y}_i \mid \boldsymbol{y}_i \in \{0,1\}^c\}_{i=1}^n$, where $c$ denotes the number of category labels.

For deep learning based multimodal approaches, we usually adopt a deep neural network to extract representation from original space into feature space. We utilize $\phi^{(j)}(\cdot)$ to denote the feature extraction function for $j$-th modality. Given data point $\boldsymbol{x}_i^{(j)}$, the feature extraction can be formed as:

$$\boldsymbol{z}_i^{(j)} = \phi^{(j)}(\boldsymbol{x}_i^{(j)}; \Phi^{(j)}),$$

where $\boldsymbol{z}_i^{(j)} \in \mathbb{R}^d$ denotes the $d$-dimension feature vector of $\boldsymbol{x}_i^{(j)}$, and $\Phi^{(j)}$ denotes the parameters of $j$-th encoder. After vectors for all modalities are extracted, we adopt a fusion function $f(\cdot)$ to fuse the different feature vectors. Then, we leverage a fully-connected layer to map the vector into $\mathbb{R}^c$. This procedure can be formed as:

$$\boldsymbol{z}_i = f(\boldsymbol{z}_i^{(1)}, \cdots, \boldsymbol{z}_i^{(m)}), \quad \hat{\boldsymbol{y}}_i = \texttt{softmax}(\boldsymbol{W}\boldsymbol{z}_i + \boldsymbol{b}).$$

Here, $\boldsymbol{W} \in \mathbb{R}^{c \times D}, \boldsymbol{b} \in \mathbb{R}^c$ denote the weights and bias of the last fully-connected layer, respectively, and $D$ denotes the dimension of $\boldsymbol{z}_i$. Then, the objective function of multimodal learning can be formulated as:

$$L_{\text{CLS}}(\boldsymbol{X}, \boldsymbol{Y}) = -\frac{1}{n} \sum_{i=1}^n \boldsymbol{y}_i^\top \log \hat{\boldsymbol{y}}_i.$$

## 3.2 Integrating Unsupervised Contrastive Learning in MML

To bridge the heterogeneous data in feature space, we utilize contrastive learning [33] in multimodal learning. For a pair of data points $\{\boldsymbol{x}_i^{(j)}, \boldsymbol{x}_k^{(l)}\}$, we define the similarity as:

$$s(\boldsymbol{x}_i^{(j)}, \boldsymbol{x}_k^{(l)}) = \frac{[\boldsymbol{z}_i^{(j)}]^\top \boldsymbol{z}_k^{(l)}}{\|\boldsymbol{z}_i^{(j)}\|_2 \|\boldsymbol{z}_k^{(l)}\|_2}.$$

The modality matching objective function can be written as:

$$L_{\text{MM}}(\boldsymbol{X}) = -\frac{1}{2n_b} \sum_i^{n_b} \Big[ \log \Big( \frac{\exp(s(\boldsymbol{x}_i^{(j)}, \boldsymbol{x}_i^{(l)})/\tau)}{\sum_k \exp(s(\boldsymbol{x}_i^{(j)}, \boldsymbol{x}_k^{(l)})/\tau)} \Big) + \log \Big( \frac{\exp(s(\boldsymbol{x}_i^{(j)}, \boldsymbol{x}_i^{(l)})/\tau)}{\sum_k \exp(s(\boldsymbol{x}_k^{(j)}, \boldsymbol{x}_i^{(l)})/\tau)} \Big) \Big],$$

where $\tau$ is the temperature parameter and $n_b$ denotes the batch size. By integrating the classification loss and modality matching loss, we can get the following objective function:

$$L_{\text{Total}} = (1 - \alpha)L_{\text{CLS}}(\boldsymbol{X}, \boldsymbol{Y}) + \alpha L_{\text{MM}}(\boldsymbol{X}), \tag{1}$$

where $\alpha$ denotes the weighted parameter between two losses.

## 3.3 Dynamic Integration

Although a fixed value of $\alpha$ allows the model to take into account both classification loss and modality matching loss, it cannot dynamically evaluate the weight of two losses during training. Hence, we propose two strategies to adjust $\alpha$ dynamically to balance two losses.

Firstly, we utilize a monotonically decreasing function to adjust the impact of category labels. The definition of the function can be written as: $\alpha_t = \omega(t)$, where $t$ denotes the number of training epochs. In this paper, we set $\omega(t) = 1 - e^{-\frac{1}{t}}$.

Then, we further exploit a learning-based integration method by utilizing bi-level optimization strategy [36]. Specifically, while considering optimizing the multimodal classification loss $L_{\text{CLS}}$, we use the minimum value of the total loss $L_{\text{Total}}$ to restrict the feasible region of the parameters $\theta$. In other words, we require the parameters not just to minimize classification loss but also to comply with a precisely defined constraint, i.e., simultaneously minimize a composite loss function—a strategically engineered combination of modality matching loss and multimodal classification loss. The specific formula is defined as follows:

$$\min_{0 \le \alpha \le 1} L_{\text{CLS}}(\theta^*(\alpha)) \quad \text{s.t.} \quad \theta^*(\alpha) \in \underset{\theta}{\arg\min} \big\{ (1 - \alpha)L_{\text{CLS}}(\theta) + \alpha L_{\text{MM}}(\theta) \big\}. \tag{2}$$

Here, $\theta$ denotes the parameters of multimodal models, and $\alpha$ emerges as a key parameter, delicately balancing modality matching loss and multimodal classification loss to direct the model toward an

**Algorithm 1:** The Proposed Algorithm.

---

**Input** : Training set $\mathcal{X}$, labels $\mathcal{Y}$, method.
**Output** : Learned parameters $\{\theta\}$ of all models.
**INIT** initialize parameters $\theta$, parameter $\alpha$, maximum iterations $T$, learning rate $\eta_\alpha$.
**for** $t = 1$ **to** $T$ **do**
    /* updating neural network parameters $\theta$. */
    **for** $i = 1$ **to** *Inner_Iters* **do**
        Calculate total loss $L_{\text{Total}}$ by forward phase.
        Update parameters $\theta$ according to its gradient.
    **end**
    /* updating weighting parameters $\alpha$ based on the chosen method. */
    **if** *method* $==$ *'learning-based'* **then**
        Calculate gradient appriximation:
            $\nabla L_{\text{CLS}}(\theta(\alpha)) = -\nabla^2_{\alpha,\theta} L_{\text{Total}} \cdot [\nabla^2_{\theta,\theta} L_{\text{Total}}]^{-1} \cdot \nabla_\theta L_{\text{CLS}}(\boldsymbol{X}, \boldsymbol{Y})$.
        Update $\alpha$ according to: $\alpha = \alpha - \eta_\alpha \nabla L_{\text{CLS}}(\theta(\alpha))$.
        Clip $\alpha$ into $[0, 1]$: $\alpha := \max(0, \min(1, \alpha))$.
    **else if** *method* $==$ *'heuristic'* **then**
        Update $\alpha$ according to: $\alpha = 1 - e^{-1/t}$.
    **end**
**end**

---

optimal balance where both types of loss are effectively managed. The optimal parameter set, $\theta^*(\alpha)$, thus represents a fine-tuned balance that, for any chosen $\alpha$, strategically minimizes this composite loss. Within Equation (2), $L_{\text{CLS}}$ is pivotal for guiding classification accuracy, while $L_{\text{MM}}$ enhances the model's ability to establish meaningful connections across different modalities.

We utilize an approximation method proposed by [16] to solve bi-level optimization problem (2) efficiently. Specifically, the gradient of $L_{\text{CLS}}(\theta(\alpha))$ with respect to $\alpha$ can be approximated by:

$$\nabla L_{\text{CLS}}(\theta(\alpha)) = -\nabla^2_{\alpha,\theta} L_{\text{Total}} [\nabla^2_{\theta,\theta} L_{\text{Total}}]^{-1} \nabla_\theta L_{\text{CLS}}(\boldsymbol{X}, \boldsymbol{Y}). \tag{3}$$

Based on the approximation Equation in (3), we can use the gradient descent method to optimize $\alpha$.

After defining the updating strategy for $\alpha$, we utilize an alternating algorithm between model parameters $\theta$ and $\alpha$ to perform model learning. Specifically, our algorithmic process iteratively refines the model parameters $\theta$ and the parameter $\alpha$, employing a nested loop structure where the inner loop focuses on the currently given $\alpha$ to minimize total loss to update $\theta$, and the outer loop updates $\alpha$ by function or bi-level policy to minimize classification losses. Through this structured optimization, the model achieves a delicate balance between multimodal matching and classification losses. The overall algorithm of our model is outlined in Algorithm (1), where we utilize $method$ to indicate the chosen updating strategy in practice. Moreover, the complexity of bi-level optimization [16] and our algorithm in Algorithm (1) is $\mathcal{O}(n)$, which makes our approach highly practical.

## 4 Experiments

### 4.1 Experimental Setup

**Datasets:** We select six widely used datasets, including KineticsSounds [2], CREMA-D [5], Sarcasm [4], Twitter2015 [49], NVGesture [42], and VGGSound [7] datasets, to validate our proposed method. Among these datasets, the KineticsSounds, CREMA-D and VGGSound datasets consist of both audio and video modalities. The KineticsSounds dataset, which contains 19,000 video clips categorized into 31 distinct actions, aims to advance video action recognition. It is divided into a training set of 15,000 clips, a validation set of 1,900 clips, and a test set of 1,900 clips. The CREMA-D dataset, encompassing 7,442 clips, is divided into six emotional categories to enhance speech emotion analysis, with 6,698 clips in the training set and 744 clips in the test set. The VGGSound dataset, which contains 310 classes and a wide range of audio events in everyday life, is a relatively large dataset. It includes 168,618 videos for training and validation, and 13,954 videos for testing. Furthermore, the Sarcasm and Twitter2015 datasets consist of image and text modalities. The Sarcasm dataset offers a compilation of 24,635 text-image pairs, divided into 19,816 for the training set, 2,410 for the validation set, and 2,409 for the test set. The Twitter2015 dataset contains 5,338

Table 1: Comparison with SOTA multimodal learning methods. The best results are highlighted in bold. The underlining symbol denotes the second best performance. The results with gray background are based on MML but perform worse than the best unimodal approach.

| Method | KineticsSounds | | CREMA-D | | Sarcasm | | Twitter2015 | | NVGesture | |
|---|---|---|---|---|---|---|---|---|---|---|
| | ACC | MAP | ACC | MAP | ACC | F1 | ACC | F1 | ACC | F1 |
| Unimodal-1 | 54.12% | 56.69% | 63.17% | 68.61% | 81.36% | 80.65% | 73.67% | 68.49% | 78.22% | 78.33% |
| Unimodal-2 | 55.62% | 58.37% | 45.83% | 58.79% | 71.81% | 70.73% | 58.63% | 43.33% | 78.63% | 78.65% |
| Unimodal-3 | – | – | – | – | – | – | – | – | 81.54% | 81.83% |
| Concat | 64.55% | 71.31% | 63.31% | 68.41% | 82.86% | 82.43% | 70.11% | 63.86% | 81.33% | 81.47% |
| Affine | 64.24% | 69.31% | 66.26% | 71.93% | 82.47% | 81.88% | 72.03% | 59.92% | 82.78% | 82.81% |
| Channel | 63.51% | 68.66% | 66.13% | 71.75% | – | – | – | – | 81.54% | 81.57% |
| ML-LSTM | 63.84% | 69.02% | 62.94% | 64.73% | 82.05% | 70.73% | 70.68% | 65.64% | 83.20% | 83.30% |
| Sum | 64.97% | 71.03% | 63.44% | 69.08% | 82.94% | 82.47% | 73.12% | 66.61% | 82.99% | 83.05% |
| Weight | 65.33% | 71.33% | 66.53% | 73.26% | 82.65% | 82.19% | 72.42% | 65.16% | 83.42% | 83.57% |
| ETMC | 65.67% | 71.19% | 65.86% | 71.34% | 83.69% | 83.23% | 73.96% | 67.39% | 83.61% | 83.69% |
| MSES | 64.71% | 72.52% | 61.56% | 66.83% | 84.18% | 83.60% | 71.84% | 66.55% | 81.12% | 81.47% |
| G-Blend | 67.12% | 71.39% | 64.65% | 68.54% | 83.35% | 82.71% | 74.35% | 68.69% | 82.99% | 83.05% |
| OGM | 66.06% | 71.44% | 66.94% | 71.73% | 83.23% | 82.66% | 74.92% | 68.74% | – | – |
| Greedy | 66.52% | 72.81% | 66.64% | 72.64% | – | – | – | – | 82.74% | 82.69% |
| DOMFN | 66.25% | 72.44% | 67.34% | 73.72% | 83.56% | 82.62% | 74.45% | 68.57% | – | – |
| MSLR | 65.91% | 71.96% | 65.46% | 71.38% | 84.23% | 83.69% | 72.52% | 64.39% | 82.86% | 82.92% |
| PMR | 66.56% | 71.93% | 66.59% | 70.36% | 83.61% | 82.49% | 74.25% | 68.62% | – | – |
| AGM | 66.02% | 72.52% | 67.07% | 73.58% | 84.28% | 83.44% | 74.83% | 69.11% | 82.78% | 82.82% |
| MLA | 70.04% | 74.13% | 79.43% | 85.72% | 84.26% | 83.48% | 73.52% | 67.13% | 83.73% | 83.87% |
| ReconBoost | 70.85% | 74.24% | 74.84% | 81.24% | 84.37% | 83.17% | 74.42% | 68.34% | 84.13% | 86.32% |
| MMPareto | 70.00% | 78.50% | 74.87% | 75.15% | 83.48% | 82.84% | 73.58% | 67.29% | 83.82% | 84.24% |
| Ours-H | 69.05% | 72.97% | 72.15% | 80.45% | 84.12% | 83.98% | 73.87% | 69.17% | 83.24% | 83.87% |
| | ±0.15% | ±0.43% | ±0.32% | ±0.85% | ±0.17% | ±0.22% | ±0.35% | ±0.26% | ±0.07% | ±0.18% |
| Ours-LB | 72.53% | 78.38% | 83.62% | 90.06% | 84.97% | 84.57% | 75.01% | 70.57% | 84.36% | 84.68% |
| | ±0.31% | ±0.37% | ±0.11% | ±1.09% | ±0.27% | ±0.18% | ±0.16% | ±0.28% | ±0.14% | ±0.24% |

text-image combinations from Twitter, with 3,179 in the training set, 1,122 in the validation set, and 1,037 in the test set. Lastly, the NVGesture dataset is used to construct research that goes beyond the limitation of two modalities. In this paper, we use RGB, Depth, and optical flow (OF) modalities for experiments, with 1,050 samples in the training set and 482 samples in the test set.

**Baselines:** We select a wide range of baselines for comparison. These baselines can be divided into two categories, i.e., traditional MML approaches and fusion methods with modal rebalancing strategies. The former category encompasses techniques like feature concatenation (CONCAT), affine transformation (Affine) [32], channel-wise fusion (Channel) [22], multi-layer LSTM fusion (ML-LSTM) [30], prediction summation (Sum), prediction weighting (Weight) [45], and enhanced trust modal combination (ETMC) [17]. And the latter category includes MSES [15], G-Blend [38], OGM [31], Greedy [41], DOMFN [46], MSLR [47], PMR [13], AGM [24], MLA [52], Recon-Boost [19], and MMPareto [40]. Other baselines including UMT [11] and QMF [51] are not adopted as these two methods have been found to be outperformed by the adopted baseline ReconBoost.

**Evaluation Metrics:** Following [31], we use accuracy (ACC) and mean Average Precision (MAP) as evaluation metrics for audio-video datasets. For text-image datasets, we adopt ACC and Macro F1-score (F1) [4]. ACC measures the proportion of correct predictions to total predictions, indicating the overall predictive accuracy. Macro F1 calculates the average of F1 scores across all categories, balancing precision and recall to evaluate performance evenly across classes. MAP represents the average precision across all categories, assessing the model's ranking ability for each category.

**Implementation Details:** In our experiments, we utilize raw data for experiments. Following [31, 13], for the KineticsSounds and CREMA-D datasets, ResNet18 [18] serves as the foundational architecture for processing both audio and video data. For video analysis, we select 10 frames from each clip and subsequently sample three frames uniformly as inputs. We adapt ResNet18's input channels from three to one to accommodate our data format [7]. In terms of audio, we transform our sound recordings into spectrograms measuring $257 \times 1004$ for KineticsSounds and $257 \times 299$ for CREMA-D,

employing the librosa [29] library for conversion. For text-image datasets, our framework incorporates ResNet50 for images and BERT [9] for text processing. We resize images to $224 \times 224$ and limit text sequences to a maximum length of 128 characters. Optimization for the audio-video datasets is conducted using stochastic gradient descent (SGD) with a momentum set to 0.9 and a weight decay parameter of $10^{-1}$. We initialize the learning rate to $10^{-2}$, progressively reducing it by a factor of ten upon observing a plateau in loss reduction, with a batch size of 256. For text-image datasets [4, 49], we employ the Adam optimizer starting with a learning rate of $10^{-4}$, with a batch size of 128. For our methods, we run the experiment three times with different random seeds and present the detailed performance with $mean$ and $std.$ values to remove randomness. All models are trained on a single RTX 3090 GPU.

## 4.2 Comparison with SOTA MML Baselines

The main results for all datasets, except VGGSound, are presented in Table 1, where "Our-H" and "Ours-LB" denote the proposed method based on heuristic strategy and learning-based strategy, respectively. In Table 1, Unimodal-1 and Unimodal-2 refer to the audio and video modalities for audio-video datasets, and the image and text modalities for image-text datasets, respectively. For NVGesture dataset, Unimodal-1/2/3 respectively denote the RGB/OF/Depth. From the results, we can derive the following observation: (1). Compared with all baselines including traditional multimodal learning approaches and fusion methods with modal rebalancing strategies, our proposed method with learning-based strategy can achieve best performance by a large margin in almost all cases. We can also find that the model with learning-based strategy can achieve better performance than that with heuristic strategy. (2). Across the Twitter2015 dataset, there is a discernible trend where the optimal unimodal performance outstrips that of multimodal joint learning. Additionally, in other datasets, fusion methodologies devoid of rebalancing mechanisms manifest negligible enhancements relative to the foremost unimodal performance, notably on the CREMA-D and Sarcasm datasets. This shortfall originates from the prevalent challenge of modal imbalance. (3). Every modality rebalancing technique demonstrates significant improvements over traditional feature concatenation fusion. This finding not only underscores the detrimental impact of modal imbalance on performance but also corroborates the efficacy of the modality rebalancing approach. (4). For NVGesture dataset, differing from modal rebalancing methods restricted to scenarios with only two modalities, such as Greedy, our approach with learning-based strategy can address challenges in scenarios involving more than two modalities and achieve best results. And our proposed method can outperform all baselines in most cases.

Table 2: Results on VGGSound dataset.

| Method | ACC | MAP |
|---|---|---|
| AGM | 47.11% | 51.98% |
| MLA | _51.65%_ | 54.73% |
| ReconBoost | 50.97% | 53.87% |
| MMPareto | 51.25% | _54.74%_ |
| Ours-H | 50.42% | 53.62% |
| Ours-LB | **52.74%** | **55.98%** |

For the relatively large dataset VGGSound, we select a set of recent algorithms, including AGM [24], MLA [52], ReconBoost [19], and MMPareto [40], for experimental evaluation. The results are shown in Table 2. From Table 2, we can see that our proposed methods can achieve the best performance in all cases compared with recent SOTA baselines on VGGSound dataset.

## 4.3 Ablation Study

To comprehensively assess the effectiveness of our proposed method, we conduct experiments to study the influence of main components, i.e., contrastive learning (CL) and dynamic integration (DI). The results are shown in Table 3, where the "CL" and "DI" denote that whether the contrastive learning and dynamic integration are applied during training. The unimodal MAP results are based on audio and video modalities for KineticsSounds and CREMA-D datasets, and unimodal F1 results are based on image and text modalities for Sarcasm and Twitter2015 datasets. Please note that dynamic integration depends on the contrastive learning loss. Hence the method with dynamic integration but without contrastive learning cannot be performed. From Table 3, we can see that both contrastive learning and dynamic integration can boost performance in multimodal learning. Moreover, by integrating contrastive learning into multimodal learning, the performance gap between audio and video is greatly reduced.

Table 3: Results of ablation study. The symbols "CL" and "DI" denote that whether the contrastive learning and dynamic integration are applied during training.

| Dataset | Module | | MAP/F1 | | | |
|---|---|---|---|---|---|---|
| | CL | DI | Multiple | Audio/Image | Video/Text | GAP |
| KineticsSounds | × | × | 69.32% | 48.82% | 27.19% | 21.63% |
| | ✓ | × | 71.76% | 51.05% | 47.05% | 3.80% |
| | ✓ | ✓ | **78.97%** | **58.40%** | **60.42%** | **2.02%** |
| CREMA-D | × | × | 76.07% | 70.97% | 34.15% | 36.82% |
| | ✓ | × | 86.32% | 72.11% | 52.51% | 19.06% |
| | ✓ | ✓ | **90.06%** | **75.27%** | **67.36%** | **7.91%** |
| Sarcasm | × | × | 82.43% | 62.81% | 77.96% | 15.15% |
| | ✓ | × | 83.10% | 68.74% | 80.72% | 11.98% |
| | ✓ | ✓ | **84.57%** | **74.53%** | **83.03%** | **8.50%** |
| Twitter2015 | × | × | 63.86% | 40.99% | 68.38% | 27.39% |
| | ✓ | × | 65.33% | 46.84% | 69.04% | 22.20% |
| | ✓ | ✓ | **70.57%** | **53.43%** | **69.68%** | **16.25%** |

Table 4: Comparison of dynamic integration strategy on KineticsSounds and CREMA-D datasets.

| Dataset | Modality | Constant | | | Stepwise | | | | Dynamic | |
|---|---|---|---|---|---|---|---|---|---|---|
| | | 0 | 0.5 | 1 | $h(0)$ | $h(1)$ | $h(0.05)$ | $h(0.95)$ | Ours-H | Ours-LB |
| KineticsSounds | Multiple | 64.55% | 64.70% | 28.67% | 65.17% | 66.92% | 66.01% | 67.41% | 69.32% | 72.89% |
| | Audio | 49.17% | 46.30% | 34.11% | 51.12% | 52.34% | 52.21% | 53.41% | 53.89% | 54.32% |
| | Video | 24.64% | 44.02% | 28.41% | 41.21% | 41.45% | 42.31% | 46.72% | 49.18% | 54.17% |
| CREMA-D | Multiple | 63.31% | 70.45% | 26.49% | 66.45% | 70.24% | 69.11% | 71.45% | 72.39% | 84.11% |
| | Audio | 55.65% | 60.17% | 33.15% | 56.19% | 57.38% | 58.09% | 60.18% | 61.89% | 65.13% |
| | Video | 18.68% | 42.54% | 20.42% | 45.14% | 49.97% | 46.41% | 55.32% | 57.14% | 64.89% |

## 4.4 Effectiveness of Integration Learning

**Analysis of Integration Strategy:** We further study the impact of different integration strategies for contrastive loss and classification loss. Specifically, we analyze three categories of integration strategy, i.e., constant, stepwise and dynamic strategy. For constant strategy, we assign a constant value to $\alpha$ for the experiment and run three sets of experiments by setting $\alpha = 0$, $\alpha = 0.5$, and $\alpha = 1$. Here, "$\alpha = 0$" denotes that we only perform supervised multimodal learning. Similarly, "$\alpha = 1$" denotes that we only perform unsupervised contrastive learning. For stepwise strategy, we define an indicator function $h(p)$, where $h(p)$ denotes that $\alpha = p$ if the current epoch is less than half of the total epochs, otherwise $\alpha = 1 - p$. For dynamic strategy, we assign value to $\alpha$ by heuristic strategy (Ours-H) and learning-based strategy (Ours-LB).

The experimental results are shown in Table 4. From Table 4, we can draw the following observations: (1). Integrating multimodal learning and contrastive learning simultaneously with a constant ratio can slightly boost performance in some cases and greatly reduce the performance gap between audio and video. (2). In general, the model with a stepwise strategy can outperform the model with a constant strategy. Furthermore, we can find that the performance of the model with two-stage training, i.e., $h(0)$ or $h(1)$, is worse than that of the model which combines two losses with a constant value, i.e., $h(0.05)$ or $h(0.95)$. (3). The overall performance of the model with the dynamic strategy is better than that with the other strategy. Moreover, the model with the learning-based strategy can achieve the best performance. And the performance gap of this model is nil or negligible. The experimental results prove that the smaller the performance gap of the uni-modals, the better the overall performance of the model.

**Change of $\alpha$ for Learning-based Strategy:** To further observe the change of the optimal $\alpha$ during training, we illustrate the change of $\alpha$ on all datasets. The results are shown in Figure 3, where $\alpha$

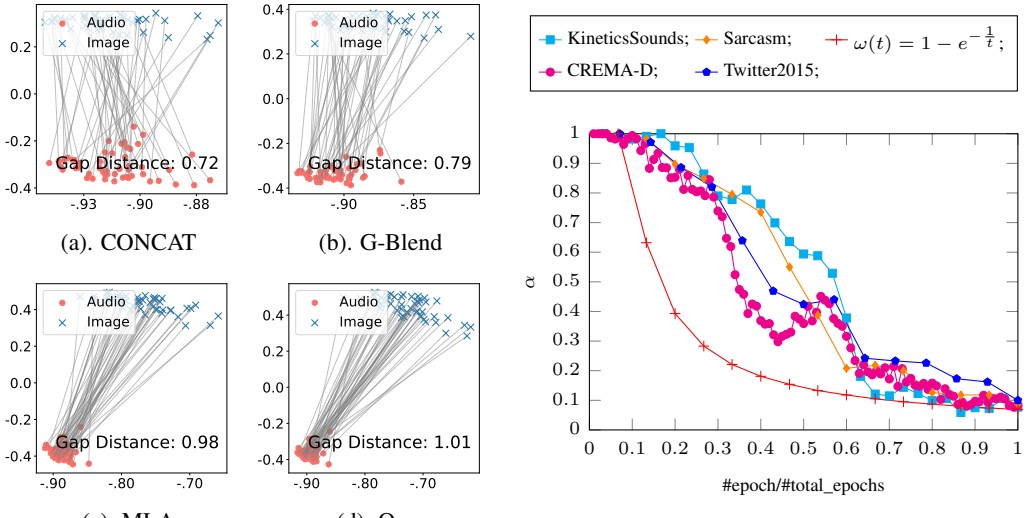

Figure 2: Visualizations of the modality gap distance on the CREMA-D dataset.

Figure 3: Change of $\alpha$ on different datasets. We illustrate the value of the heuristic integration strategy for comparison.

is calculated by learning-based strategy, and $\omega(t)$ denotes the heuristic based strategy. As the total epochs for different datasets are different, we change the x-axis as the proportion of the current epoch, i.e., #epoch/#total_epochs. From Figure 3, we can draw the following observations: (1). The general trend of the change for $\alpha$ is roughly the same on different datasets. (2). The customized function $\omega(t)$ is close to the actual changes to some extent, but there is still a gap between the customized function and the actual changes. In practice, it is difficult to fit the change of parameter $\alpha$ perfectly. Hence, we can see that our method has good adaptability in different scenarios. Furthermore, by comparing the trend of $\omega(t)$ with other curves, a natural question arises: if the curve of the heuristic algorithm aligns more closely with the trend of the change curve based on the learning-based strategy, will the effect be better? The answer is yes. This issue can be easily verified by using a third-order polynomial function to approximate the learning-based curve. Specifically, by substituting the $\omega(t)$ as the polynomial function $\hat{\omega}(t) = at^3 + bt^2 + ct + d$, where $a = 1.5 \times 10^{-4}, b = -6.5 \times 10^{-3}, c = 3.2 \times 10^{-2}, d = 1$, we can achieve higher accuracy with $71.22\%$ and MAP with $76.28\%$ on KineticsSounds dataset compared with ours-H. However, this strategy can only be applied once the actual changing trend of $\alpha$ is observed.

## 4.5 Further Analysis

**Analysis of Modality Gap:** As mentioned in the paper [25], modality gap characterizes the correlation between different modalities in multimodal learning. And large modality gap leads to better performance in some situations. We further illustrate the modality gap for CONCAT, G-Blend, MLA, and our method. The results are shown in Figure 2. From Figure 2, we can find that our method can learn more discriminative representations and results in higher accuracy with a large modality gap compared with other methods.

**Robustness Analysis of the Pretrained Model:** We further exploit the robustness of the large-scale language vision pretrained model CLIP [33] model on Sarcasm and Twitter2015 datasets. We replace the encoders for image and text as the corresponding encoders pretrained by CLIP and fine-tune the model on Sarcasm and Twitter2015 datasets respectively. The results are shown in Table 5, where "CLIP+MLA" and "CLIP+Ours" present that we apply the MLA's and ours algorithm, respectively. From Table 5, we can draw the following observations: (1). Both CLIP+MLA and CLIP+Ours can outperform CLIP in all cases. (2). With the help of dynamic integration, the performance of our method is better than that of MLA.

**Visualization:** We utilize GradCAM [34] to showcase the visualization of image regions that attract the weak modality's focus during training. By using GradCAM, the importance scores are assigned

Table 5: Results on the Sarcasm and Twitter2015 datasets achieved by using the CLIP pre-trained model as encoders.

| Method | Sarcasm | | | Twitter2015 | | |
|---|---|---|---|---|---|---|
| | Image | Text | Multiple | Image | Text | Multiple |
| CLIP | 74.82% | 82.15% | 83.11% | 54.48% | 71.75% | 72.52% |
| CLIP+MLA | 77.45% | 83.19% | 84.45% | 56.53% | 72.37% | 73.95% |
| CLIP+Ours | **79.78%** | **83.67%** | **85.42%** | **64.67%** | **72.59%** | **74.43%** |

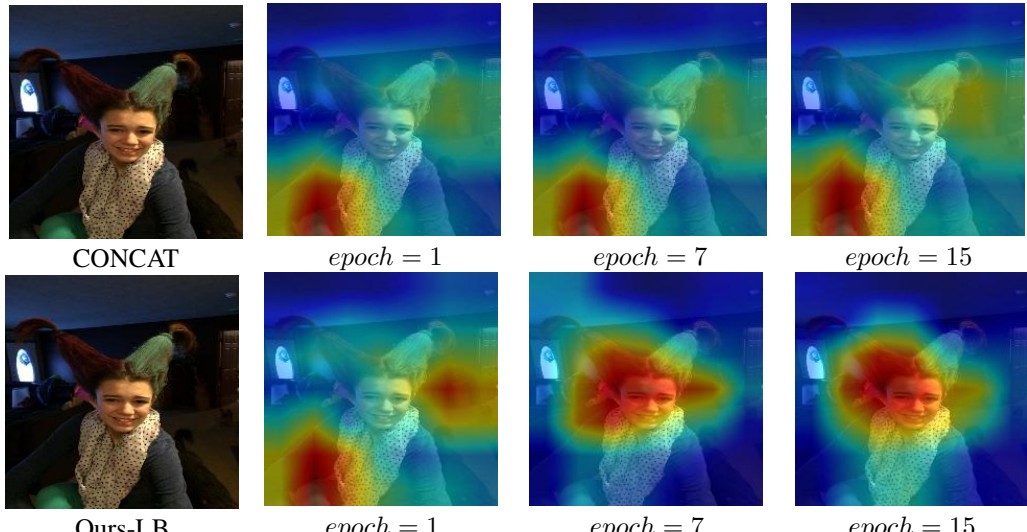

Figure 4: Visualization on Twitter2015 dataset. Our proposed method tends to perform feature learning first and then fit the learned features to the category labels.

to every pixel in each feature map, aiding in identifying the image regions critical for the model's predictions. We compare the visualization for CONCAT and our proposed method. The visualization results are presented in Figure 4, where the second, the third, and the last columns denote the results of the first, the seventh, and the last epoch, respectively. The category label for this image is "Negative" and the corresponding text is "Crazy hair day ! $T$ is a contender.". By comparing our method with CONCAT, we can see that our method focuses on the textual information from text modality, and then fits the learned features to the category labels.

## 5 Conclusion

In this paper, we discuss a core reason for modality imbalance in multimodal learning, i.e., fitting category labels. We find that appropriate positive intervention label fitting can correct the difference in learning ability for different modalities, thus alleviating the modality imbalance phenomenon. Based on this observation, we propose a novel multimodal learning approach to overcome modality imbalance problem by dynamically integrating unsupervised contrastive learning and supervised multimodal learning. We design a heuristic strategy and a learning based strategy to perform integration dynamically. Experiments on various datasets demonstrate that our method can boost performance in multimodal learning.

**Limitations:** For the limitations of our proposed method, the root cause of modality imbalance phenomenon caused by fitting category labels is worth discussing in depth. Does the specific category label contain attributes that are more suitable for fitting a certain modality? We leave it as a future work.

## Acknowledgments

The authors thank the anonymous reviewers for their valuable comments. This work is partially supported by National Key RD Program of China (2022YFF0712100), NSFC (62276131), Natural Science Foundation of Jiangsu Province of China under Grant (BK20240081), the Fundamental Research Funds for the Central Universities (No.30922010317).

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
