# OpenReview forum: "Facilitating Multimodal Classification via Dynamically Learning Modality Gap"
_NeurIPS.cc/2024/Conference — NeurIPS 2024 poster_

### Official Review · Reviewer_XdY4 · 2024-07-06

**Soundness:** 1
**Presentation:** 3
**Contribution:** 1
**Rating:** 3
**Confidence:** 4

**Summary:**

This paper proposes a novel multimodal learning method to address modality imbalance by dynamically integrating unsupervised contrastive learning and supervised multimodal learning. The authors design two dynamic integration strategies: heuristic and learning-based. Experimental results show that the proposed method significantly outperforms existing multimodal learning methods on several datasets.

**Strengths:**

- Clarity. The presentation of this paper is generally expectational. I enjoy the clarity of writing.
- Important topics. Multimodal learning and modality imbalance have achieved increasing research interest recently.
- Experimental results show that the proposed method significantly outperforms existing multimodal learning methods on several datasets.

**Weaknesses:**

**Weaknesses:**

1. **Insufficient Analysis of Modality Imbalance**: The paper claims to be the first to analyze modality imbalance from the perspective of fitting category labels. However, only a simple experiment on the Kinetics Sounds dataset is presented, with an inadequate analysis. The experiment uses uniform labels and one-hot labels to obtain a mixed loss, finding that increasing the weight of LUL_ULU reduces the modality performance gap. This seems obvious, but how this phenomenon relates to the claim that "this also implies that fitting category labels is a core cause of modality imbalance in multimodal learning" is not clearly explained.
2. **Difference Between Mixed Loss and Label Smoothing**: What is the difference between the so-called mixed loss and the label smoothing technique? The authors should clarify this distinction. Label smoothing first obtains mixed labels and then calculates the loss. In the context of cross-entropy, these should be equivalent. It would be more convincing if the authors' analysis could relate to the theoretical explanations of label smoothing.
3. **Motivation and Method Inconsistency**: The paper's motivation is to alleviate multimodal imbalance by intervening in fitting category labels. However, it then introduces contrastive loss to intervene in the feature space. There is insufficient connection between the method and the motivation. The explanation "Ideally, contrastive learning can also mitigate the effect of modality imbalance problem. Hence, we introduce contrastive learning to impose positive intervention in multimodal learning to alleviate the impact of fitting labels." is too superficial. The authors should provide a deeper and more solid analysis of how contrastive learning helps in intervening in label fitting.
4. **Practicality of Dynamic Weight Methods**: Section 3.3 introduces heuristic and learning-based dynamic weights. The heuristic dynamic weights do not perform well in experiments; the learning-based dynamic weights involve bi-level optimization. Although the authors used an approximation method to derive Equation (3), it still involves calculating the Hessian matrix and its inverse. The authors did not discuss the computational cost or approximation techniques, making the practicality of this method questionable.

**Questions:**

Please refer to weakness and address my concerns.

---

> ### Author Rebuttal · Authors · 2024-08-06
>
> Thanks for your comments.\
> **Overall Response:**  Our motivation is to explore the relationship between label fitting (one-hot label) and modality imbalance, thus to find a way to mitigate the impact of label fitting. We first utilize a toy noise label experiment to demonstrate that label fitting can lead to modality imbalance. Specifically, we compared the classification performance by using the original one-hot label loss ($L_s$), a label free loss ($L_u$) and their linear combination loss. One can image that $L_s$ is a supervised loss while $L_u$ is a unsupervised loss. The results in Figure 1 of the original paper show that even the overall classification performance of $L_u$ is worse than that of $L_s$, $L_u$ has the smallest modality performance gap, indicating that the unsupervised loss could mitigate the modality imbalance but lead to low classification performance. This inspires that there may be an optimal trade-off between the supervised loss $L_s$ and unsupervised loss $L_u$. To this end, one simple way is to tune the $\alpha\in [0,1]$ for $\alpha L_s + (1-\alpha)L_u$. We finally found that $0.7L_s+0.3L_u$ has the best overall classification performance with the middle modality performance gap, compared with $L_s$ and $L_u$. Based on the experimental findings, we then employ the contrastive unsupervised learning to align the modality representation of multimodal data [1], thereby mitigating the modality imbalance. In this way, overall classification performance of MML can be improved. It is worth to mention that the proposed method is a generic algorithm that can be integrated with many unsupervised learning.\
> **1. Answer for Question "Insufficient Analysis of Modality Imbalance":** The overall classification performance of MML is a primary concern to us and is affected by the modality imbalance phenomenon. Literature [2,3,4] theoretically reveals the intrinsic relationship between modality performance gap and modality imbalance, i.e., the larger the modality performance gap, the more serious the modality imbalance. Based on these theoretical conclusions and our findings, we deeply explore the impact of label fitting on modality imbalance in this paper. The main logic of our paper is that label fitting leads to the modality performance gap, which in turn exacerbates the modality imbalance phenomenon. As you mentioned, increasing the weight of noise labels to reduce the impact of label fitting is positively correlated with a reduction in the modality performance gap. Meanwhile, as the modality performance gap narrows, the modality competition phenomenon will be alleviated [3], thus alleviating the modality imbalance. In summary, the overall performance of MML is improved through intervening the label fitting.\
> **2. Answer for Question "Difference Between Mixed Loss and Label Smoothing":** The mixed loss is the label smoothing. As mentioned in the overall response part, the experiment of label noise aims to explore the relationship between label fitting and modality performance gap, thereby affecting overall performance. Therefore, we did not discuss the mixed loss and label smoothing in the paper. We will add appropriate discussion to the final version.\
> **3. Answer for Question "Practicality of Dynamic Weight Methods":** Heuristic method is designed primarily to validate the motivation behind our methods. For example, by carefully adjusting the function form, such as using the polynomial function $f(t) = a t^3 + b t^2 + c t + d$, with coefficients $a = 1.5 \times 10^{-4}$, $b = -6.5 \times 10^{-3}$, $c = 3.2 \times 10^{-2}$, $d = 1$, the classification accuracy achieves 71.22% compared to 70.04% with the MLA on the Kinetics-Sounds dataset. \
> In fact, the complexity of bi-level optimization [5] and our algorithm is O(n). More details of bi-level optimization can be referred to literature [5]. To further demonstrate the efficiency of our method, we adopt MLA and our method for experiments and report the training time cost on Kinetics-Sounds dataset. The average time costs for MLA and our method are 327s and 301s for an epoch, respectively. Therefore, we can demonstrate the practicality of our method. We will add some discussion and results in the final version.\
> **References:**\
> [1]. Fan, Yunfeng, et al. Detached and Interactive Multimodal Learning. ACMMM, 2024.\
> [2]. Zhang, Xiaohui, et al. Multimodal Representation Learning by Alternating Unimodal Adaptation. CVPR, 2024.\
> [3]. Huang, Yu, et al. Modality competition: What makes joint training of multi-modal network fail in deep learning? (provably). ICML. 2022.\
> [4]. Wang, Wei, and Zhi-Hua Zhou. Co-training with insufficient views. ACML. 2013.\
> [5]. Vicente, Luis N., and Paul H. Calamai. Bilevel and multilevel programming: A bibliography review. JGO, 1994.

---

> > ### Author Response · Authors · 2024-08-13
> > **Please Review Rebuttal Response**
> >
> > We want to kindly remind you that our rebuttal is available for your review. If you have any further questions or require additional clarification, please feel free to reach out to us directly. Thank you very much for your time and consideration.

---

> > > ### Comment · Reviewer_XdY4 · 2024-08-13
> > >
> > > I appreciate the efforts made by the authors during the rebuttal. However, my main concern remains with Question 3, i.e., Motivation and Method Inconsistency, has not been adequately addressed. The authors showed that intervening in label fitting leads to a reduction in the performance gap between modalities, but also results in overall performance degradation. Consequently, the authors concluded that intervening in label fitting can reduce the differences between modalities in the feature space. Thus, the authors introduced contrastive loss, claiming that it was intended to narrow the modality gap by intervening in the labels. However, this logic is hard for me to understand without solid theoretical analysis. I am keeping my original score.

---

> > > > ### Author Response · Authors · 2024-08-13
> > > > **Response for Comments.**
> > > >
> > > > Thanks for your response.
> > > >
> > > > We regret that your concerns have not been fully addressed.
> > > > Our logic is not “intervening in label fitting can reduce the differences between modalities in the feature space. **Thus**, the authors introduced contrastive loss, claiming that it was intended to narrow the modality gap by intervening in the labels”. Intervening in label fitting is just one reason to introduce contrastive learning. More importantly, contrastive learning aims to learn similar representations for data pairs across different modalities [1], thereby aligning the multimodal representations. Hence, we utilize contrastive learning to intervene in label fitting, as we metioned in our response to reviewer g7cu. In other words, contrastive learning is chosen for two reasons, not just one.
> > > >
> > > > Let us explain this from another perspective. Both noise label and contrastive learning can intervene in label fitting. However, we find that noise label leads to overall performance degradation, which contradicts our original motivation. In contrast, contrastive learning effectively achieves multimodal feature alignment [1]. Hence, we utilize contrastive learning to intervene the label fitting. Furthermore, this intervention is dynamic as the learning process. In the initial stage, the model primarily focuses on feature learning. As training progresses, label fitting becomes increasingly important. The experiments also verify this viewpoint.
> > > >
> > > > Regarding the theoretical analysis you mentioned, we kindly request that you elaborate on the specific concerns or the particular aspect of theoretical analysis you would like to see. We believe this would greatly assist us in improving our work.
> > > >
> > > > We look forward to hearing your valuable response to our comments.
> > > >
> > > > [1]. Fan, Yunfeng, et al. Detached and Interactive Multimodal Learning. ACMMM, 2024.

---

### Official Review · Reviewer_CoWQ · 2024-07-11

**Soundness:** 3
**Presentation:** 3
**Contribution:** 3
**Rating:** 7
**Confidence:** 5

**Summary:**

This paper proposes a novel multimodal learning approach by integrating supervised MML and unsupervised contrastive learning dynamically to address the multimodal imbalance problem. The authors begin by examining the differences in unimodal performance, analyze the reasons for these differences, and introduce unsupervised learning to eliminate them, thereby solving the problem of modality imbalance. Experiments on widely used datasets clearly demonstrate the superiority of the proposed method. The approach of analyzing modality imbalance from the label fitting perspective is particularly novel. The authors conducted comprehensive experiments to prove the effectiveness of the proposed method.

**Strengths:**

1. The approach of analyzing modality imbalance from the label-fitting perspective is particularly novel. Most of the existing multimodal learning approaches focus on designing adaptive learning strategies. This paper analyzes the causes of modality imbalance from another perspective and provides some inspiration for the development of multimodal learning.
2. The paper is well organized and the writing is easy to follow. The overall logic of the paper is clear.
3. To demonstrate the effectiveness of the proposed method, the authors provide detailed explanations from both methodological and experimental perspectives. Compared with SOTAs, we can find that the proposed method can eliminate the modality imbalance phenomenon and boost performance.
4. The authors design a heuristic and learning-based strategy to integrate two losses. Furthermore, the authors analyze the effects of fixed value strategies (constant), heuristic strategy, and learning-based from an experimental perspective.

**Weaknesses:**

1. According to Figure 3, if a function aligns more closely with the trend of the learning strategy curve than \omega(t), would it perform better than \omega(t)?

**Questions:**

Please refer to weakness.

**Limitations:**

Yes.

---

> ### Author Rebuttal · Authors · 2024-08-06
>
> Thank you for your comments.\
> **1.Answer for Question "Closely With the Learning Strategy Curve"**: We use the polynomial function $f(t) = a t^3 + b t^2 + c t + d$, with coefficients $a = 1.5 \times 10^{-4}$, $b = -6.5 \times 10^{-3}$, $c = 3.2 \times 10^{-2}$, $d = 1$ to fit the current learning curve and represent  $\omega(t)$. The results in Table 1 indicate some degree of improvement. However, the functions fitted to different datasets are different, and it is difficult to find a unified function form, so we put forward the learning-based method.
>
> Table 1. Performance of different weights on Kinetics-Sounds.
>
> |Method|ACC|MAP|
> |:---|:---|:---|
> |Ours-H|	69.05|	72.97|
> |$a t^3 + b t^2 + c t + d$|	71.22|	76.82|
> |Ours-LB|	**72.53**|	**78.38**|

---

> > ### Comment · Reviewer_CoWQ · 2024-08-10
> >
> > Thank you for the helpful response that addressed my concern.

---

### Official Review · Reviewer_hRBu · 2024-07-12

**Soundness:** 3
**Presentation:** 2
**Contribution:** 3
**Rating:** 6
**Confidence:** 5

**Summary:**

This paper discusses a core reason for modality imbalance in multimodal learning, i.e., fitting category labels. It proposes a novel multimodal learning approach to overcome the modality imbalance problem by dynamically integrating unsupervised contrastive learning and supervised 319 multimodal learning.

**Strengths:**

This paper reveals the reason that models converge at different rates is the difficulty of fitting category labels is inconsistent for each modality during learning. This is a novel insight into imbalanced multimodal learning.

This paper further proposes a novel multimodal learning approach by integrating unsupervised contrastive learning and supervised multimodal learning.

**Weaknesses:**

1. The reviewer is curious about the performance of the proposed method on larger datasets, such as VGGsound.

2. When the information of two modalities is unbalanced, it is optimal to rely more on the modality that contains more information. However,  the proposed method is forcibly narrowing the modality gap. Will this lead to the restriction of the learning of strong modality and lead to sub-optimal performance?

**Questions:**

See weakness

**Limitations:**

Yes, they do.

---

> ### Author Rebuttal · Authors · 2024-08-06
>
> Thank you for your detailed comments.\
> **1.Answer for Question "Performance on Larger Dataset"**: We compare our method with the state-of-the-art method MLA on the VGGSound dataset using the experimental settings described in the literature [1]. Experimental results in Table 1 demonstrate that our method achieves higher accuracy on larger datasets compared to state-of-the-art method MLA.
>
> Table 1. Performance comparison between state-of-the-art method and our method on VGGSound dataset.
> |Method|Acc|MAP|
> |---|---|---|
> |MLA|51.65|54.73|
> |Ours|**52.74**|**55.98**|
>
> **2. Answer for Question "Restriction of Strong Modality"**: Table 4 in the original paper shows that, compared to the standard joint training method, both the strong and weak modalities have shown improvement, with the weak modality improving more than the strong modality. Additionally, the single-modality performance of combined training is close to that of individual training.\
> **References:**\
> [1]. On Uni-Modal Feature Learning in Supervised Multi-Modal Learning. ICML 2023.

---

> > ### Comment · Reviewer_hRBu · 2024-08-12
> >
> > Thank you for the helpful response that addressed my concern. i will raise my score to weak accept

---

> > > ### Author Response · Authors · 2024-08-13
> > >
> > > Thank you very much for your positive feedback. I'm pleased that my response effectively addressed your concerns.

---

### Official Review · Reviewer_g7cu · 2024-07-12

**Soundness:** 2
**Presentation:** 3
**Contribution:** 2
**Rating:** 7
**Confidence:** 5

**Summary:**

This paper focuses on the notorious multi-modal imbalance problem. The authors attribute this issue to different modality-specific convergence rates caused by the inconsistent difficulty of fitting class labels. Therefore, this paper adopts unsupervised contrastive learning to mitigate the modality gap. Moreover, the learning-based integration strategy is proposed to harmonize unsupervised and supervised optimization objectives. Experiments demonstrate the efficacy.

**Strengths:**

This paper explores an interesting problem. In practice, the dominant modality usually overpowers the overall learning process, and how to pursue reconciliation between different modalities is critical but challenging in multi-modal learning.

The observation that introducing random noise into accurate one-hot labels can alleviate modality imbalance is novel and clearly illustrated.

The proposed method is presented clearly and easy to reproduce.

**Weaknesses:**

**The motivation for using contrastive learning to mitigate modality imbalance is not reasonable.** The authors' view that different patterns from the same sample should be as close as possible in the feature space seems incorrect. An ideal multi-modal representation should capture both shared and unique information relevant to the downstream task [1,2,3], **which are not necessarily identical**. The authors should explain in depth why introducing contrastive learning can mitigate modality imbalance and achieve better representation.

**The related work section lacks discussion on recent SOTA research.** QMF [4] provides a quality-aware multimodal fusion framework to mitigate the influence of low-quality multimodal data. ReconBoost [3] finds that the major issue arises from the current joint learning paradigm. The proposed alternating learning paradigm pursues a reconciliation between different modalities and achieves significant improvements. MMPareto [5] resolves conflicts between multi-modal and uni-modal gradients under multi-modal scenarios and effectively alleviates modality imbalance.

**The experiments lack comparison with the UMT, QMF, ReconBoost, and MMPareto.** Additionally, the authors should evaluate the performance of the uni-modal encoder to further demonstrate that the proposed method can alleviate modality imbalance.

[1] On Uni-Modal Feature Learning in Supervised Multi-Modal Learning. ICML 2023.

[2] Factorized Contrastive Learning: Going Beyond Multi-view Redundancy. NeurIPS 2023.

[3] ReconBoost: Boosting Can Achieve Modality Reconcilement. ICML 2024.

[4] Provable Dynamic Fusion for Low-Quality Multimodal Data. ICML2023

[5] MMPareto: Boosting Multimodal Learning with Innocent Unimodal Assistance. ICML 2024

**Questions:**

Additionally, in table5, the authors compare the results using the CLIP pre-trained encoder. I am concerned about how to measure CLIP's performance on a single modality. Should a classifier head be fine-tuned for this purpose? The authors should give implementation details about Tab.5.

---

> ### Author Rebuttal · Authors · 2024-08-06
>
> Thank you for your comments. \
> **1. Answer for Question "Contrastive Learning can Mitigate Modality Imbalance"**: Our motivation aims to find a way to mitigate the modality performance gap caused by label fitting. As shown in literature [1], different modalities will compete with each other during the joint-training. Specifically, the neural network will not efficiently learn all features from different modalities, and only a subset of modality encoders will capture sufficient feature representations. Meanwhile, contrastive learning aims to learn similar representations for data pairs of different modalities [2], thus aligning the multimodal representations for all modalities. Hence, we utilize the contrastive learning to intervene the label fitting. Furthermore, this intervention is dynamic as the learning process. In the initial stage, the model primarily focuses on feature learning. As training progresses, label fitting becomes increasingly important. This phenomenon aligns with our expectations.\
> **2. Answer for Question "Lack Comparison with SOTA Method"**: We conduct a comparison with existing state-of-the-art methods. The results in Table 1 demonstrate that our method achieves higher classification accuracy compared to these methods. Notably, our approach significantly narrows the modality performance gap, which contributes to an improvement in overall classification performance. We will add more detailed analysis in the final version.
>
> Table 1. Performance comparison between state-of-the-art methods and our method on Kinetics-Sounds
> |Method | Acc | Audio Acc | Video Acc | Modality Performance Gap |
> |---|---|---|---|---|
> |UMT[3] | 66.10 | 54.35 | 45.30 | 9.05 |
> |QMF[4] | 65.78 | 51.57 | 32.19 | 19.38 |
> |ReconBoost[5] | 70.85 | 53.79 | 52.12 | 1.67 |
> |MMPateto[6] | 70.13 | **56.40** | 53.05 | 3.35 |
> |Ours | **73.17** | 54.07 | **53.49** | **0.58** |
>
> **3. Answer for Question "Results using CLIP Pre-trained Encoder"**: In this case, following the setting of [7], we use the pre-trained weights in the CLIP as initialization. In the subsequent training process, the feature extractor and classification head are fine-tuned. We will add more details in the final version.\
> **References:**\
> [1]. Huang, Yu, et al. Modality competition: What makes joint training of multi-modal network fail in deep learning?(provably). ICML. 2022.\
> [2]. Fan, Yunfeng, et al. Detached and Interactive Multimodal Learning. ACMMM, 2024.\
> [3]. Du, Chenzhuang, et al. On Uni-Modal Feature Learning in Supervised Multi-Modal Learning. ICML 2023.\
> [4]. Hua, Cong, et al. ReconBoost: Boosting Can Achieve Modality Reconcilement. ICML 2024.\
> [5]. Zhang, Qingyang, et al. Provable Dynamic Fusion for Low-Quality Multimodal Data. ICML 2023.\
> [6]. Wei, Yake, et al. MMPareto: Boosting Multimodal Learning with Innocent Unimodal Assistance. ICML 2024.\
> [7]. Zhang, Xiaohui, et al. Multimodal Representation Learning by Alternating Unimodal Adaptation. CVPR, 2024.

---

> > ### Comment · Reviewer_g7cu · 2024-08-08
> > **Response for Rebuttals.**
> >
> > Thank you for your excellent efforts during the rebuttal. The authors have addressed most of my concerns, and now I believe this is a qualified manuscript for publication. I have raised my score to accept it.

---

> > > ### Author Response · Authors · 2024-08-09
> > >
> > > Thank you for your valuable feedback and for acknowledging our work.

---

### Decision · Program_Chairs · 2024-09-25

**Decision:**

Accept (poster)

**Comment:**

The paper proposes a novel multimodal learning approach to overcome the modality imbalance problem. It received overall positive reviews and the reviewers appreciated the problem addressed, proposed framework and writing. One of the reviewers still has concerns regarding the inconsistency between the motivation and the proposed framework. The authors have tried to explain it, but no theoretical justification has been provided. Based on the overall positive reviews, the recommendation is to accept the paper. The authors are strongly encouraged to address any remaining concerns in the camera-ready version.